# Nursing perspectives on advancing language access in the emergency department: A qualitative study

Daniel Cordova [1,2]*, Jesus R. Torres[2,3,4,5], Shiou Udagawa[3], Xin Qi[2], Tucker Avra[2], Breena R. Taira[2,4]

1 Charles R. Drew University of Medicine and Science, Los Angeles, California, United States of America, 2 UCLA David Geffen School of Medicine, Los Angeles, California, United States of America, 3 Department of Emergency Medicine, UCLA, Los Angeles, California, United States of America, 4 Department of Emergency Medicine, Olive View-UCLA Medical Center, Sylmar, California, United States of America, 5 National Clinician Scholars Program at UCLA, Los Angeles, California, United States of America

* Dcordova3@dhs.lacounty.gov

## Abstract

### Objective

Failure to provide language services in health care settings negatively impacts patients with non-English language preferences (NELPs), yet underuse of language assistance remains rampant. Nurses in the Emergency Department (ED) handle critical communication in a fast-paced environment, posing a communication challenge. The aim of this study is to describe ED nursing perspectives on barriers and facilitators to accessing language services to construct interventions that increase the uptake of language services when interacting with ED patients with NELPs.

### Methods

This is a qualitative study of ED nursing staff at two sites using the Behavior Change Wheel (BCW). Participants included registered nurses, nurse practitioners, and nursing assistants. The BCW was used to identify barriers and facilitators to accessing formal language services (professional interpreters, remote or in-person) which were mapped to intervention functions to construct proposed interventions.

### Results

A total of 36 interviews were conducted with registered nurses (n = 29), nurse assistants (n = 3), and nurse practitioners (n = 4). Barriers and facilitators to calling an interpreter were identified in all three of the BCW categories of capability, opportunity and motivation. These were mapped to intervention functions on the BCW to construct tangible interventions including restructuring the environment to have designated parking spaces for video remote interpreter machines in private areas, standardized

**Data availability statement:** All relevant data are within the manuscript and its supporting information files as this is a qualitative study. Also, providing the full qualitative transcripts would increase the risk that the participant could be identified.

**Funding:** DC: UCLA David Geffen School of Medicine Global Health Pathway Short Term Training Program JT: National Clinician Scholars Program at UCLA https://nationalcsp. org/ Funders played no role in study design, data collection and analysis, decision to publish or preparation of the manuscript.

**Competing interests:** The authors have declared that no competing interests exist.

training on language access services, equipment and policies, and training senior nursing language access champions to model the behavior of calling interpreters and to discourage ad hoc interpretation.

## Conclusion

We identified nursing perspectives on barriers and facilitators to language access and used the BCW to construct interventions. ED administrators can use these interventions as part of efforts to eliminate the underuse of language services and the potential negative impact outcomes for patients with a NELP.

## Background

Patients with a non-English language preference (NELP) are common in clinical settings. Lack of access to interpreter services in healthcare contributes to persistent population level health disparities that negatively impact immigrant communities [1]. More than 25 million people in the United States speak English less than very well [2]. This is magnified in settings with diverse patient populations such as Los Angeles [3]. Providing language concordant care improves healthcare outcomes, and prevents errors from miscommunication [4–6]. Proper communication and understanding also helps reduce the use of overly aggressive diagnostics in situations where more limited testing may suffice [7]. Previous studies have demonstrated the clinical benefit of conducting care in a patient's preferred language [8–11]. There is also a legal framework that entitles patients to language assistance [12]. Despite this, there remains an underuse of interpreters in clinical settings and patients with a NELP frequently leave medical encounters with limited understanding of what has transpired [8].

The Emergency Department (ED) is a dynamic environment where many critical conversations occur with patients with NELPs, however, historically, language access in the ED has been suboptimal [13,14]. Nurses are integral members of the ED team and are responsible for multiple areas of critical patient communications including triage, medication reconciliation and discharge instructions. Understanding how nursing staff utilizes language services in the ED is important considering their time sensitive and varied responsibilities. Despite being charged with such important communication, barriers and facilitators to working with interpreters have not been described from the perspective of ED nursing staff. Nursing perspectives are therefore key to optimizing the chances of success when designing ED-based interventions for improving rates of language access for patients with a NELP.

### Objective

The aim of this study is to understand the perspectives of nursing staff regarding language access for NELP patients in the ED. We use qualitative interview data and apply the Behavior Change Wheel (BCW) framework to design viable interventions and implementation strategies to improve language access in the ED.

## Methods

This is a qualitative study using an implementation science approach consisting of semi-structured interviews of Emergency Department (ED) nursing staff to understand their perspectives on language access for patients with a NELP in the ED.

### Theoretical framework

We used an implementation science approach and chose the BCW as the theoretical framework. When approaching a clinical problem with the goal of changing a provider behavior (such as calling an interpreter) an implementation science approach can support development of an intervention that has the highest likelihood of achieving an optimal result [15]. Implementation science is defined as "the scientific study of methods to promote the systematic uptake of research findings and other evidence practices into routine practice" [16,17]. The BCW framework, with the COM-B theory of behavior change at its core, was chosen *a priori* to assist in mapping qualitative findings to intervention functions and thus optimize the chances that suggested interventions would increase the rate of interpreter use. The BCW has three stages of development of a behavior change intervention: understanding the behavior, recognizing intervention options; and identifying the appropriate behavior change techniques [18]. The BCW uses the COM-B model of behavior where the target behavior (B) is assumed to be the result of the interactions between an individual's capability (C), opportunity (O), and motivation (M) which can be subdivided into physical and psychological capability, physical and social opportunity, and automatic and reflective motivation. For a visual of the BCW and its COM-B components, we encourage the reader to visit the authors' website [19]. The target behavior of interest was defined as a member of the nursing staff calling an interpreter during communication with a patient with a NELP.

### Context

Interviews took place in the ED's of Olive View-UCLA Medical Center (OVMC) and Ronald Regan UCLA Medical Center (RRMC). OVMC is an urban safety-net hospital with an academic affiliation and an annual census of approximately 60,000 patients. RRMC is an academic quaternary referral center with an annual census of 50,000 patients. At the time of the study, at OVMC there were phones that direct-dialed the remote phone-based interpreter in every patient room, video remote interpreter (VRI) machines, and one in-person Spanish language interpreter weekdays from 8am to 11 pm. There is a bilingual certification program for staff sponsored by the health system. At RRMC there were VRI machines and a phone line but no in-person interpreters or bilingual certification program for staff.

### Participants and sampling

Purposive sampling was used to identify participants from each of the following categories: nurse practitioners (NPs), nurses (RNs), and nursing assistants (NAs). All nursing staff in both EDs were eligible and participants were recruited in-person in the ED and via email. Interviews continued until thematic saturation was reached after discussion and agreement of the research team defined as the absence of new codes or themes in participant responses [20]. This study was determined to be exempt by the Olive View-UCLA Education Research Institute Institutional Review Board (IRB) and a waiver of written informed consent was granted. The UCLA Office of the Human Research Protection Program also approved the project prior to the commencement of any research. Verbal consent was obtained from all participants prior to starting any interview.

### Data collection procedures

Recruitment took place from May 1, 2021 to March 1, 2024. One-time interviews were offered as in person or by telephone with a planned duration of 30–45 minutes. In person interviews were conducted during staff break times and before or after shifts within the ED in a private space away from any active clinical areas. Telephone interviews were conducted

at the participants' convenience during non-work hours. Verbal consent was obtained and semi structured interviews were completed based on a predetermined interview guide that had been trialed for understandability. All enrolled participants completed the study. Interviews topics included general perceptions of language barriers, how staff utilize existing language services, and how language services could be optimized. We also collected demographics including position, languages spoken, and years of experience. All interviews were audio-recorded, transcribed and checked for accuracy by the study team. Participants did not have the opportunity to review transcripts of their conversations. No field notes were taken.

## Qualitative analysis

Qualitative data were anonymized and transcribed verbatim using a professional online transcription service. Transcripts were reviewed to ensure accuracy. Analysis was conducted using ATLAS.ti (Berlin, Germany) a collaborative, cloud-based qualitative data analysis platform. A hybrid deductive/inductive thematic analysis approach was used with the BCW as an initial framework for coding but leaving space for salient inductive codes that fell outside the framework [21]. The qualitative researchers (DC, TA, SU, JT, BT) met regularly during the coding process to discuss the coding, refine, and consolidate identified themes, and resolve coding discrepancies. Member checking and deviant case analysis were used to enhance trustworthiness. Identified themes were then categorized as either a barrier or facilitator to the target behavior of a provider calling for an interpreter, and barriers and suggestions for optimization were mapped to intervention functions using the BCW. Intervention functions were used to suggest interventions with the most potential to impact behavior.

## Researcher characteristics and reflexivity

The research team was made up of emergency physicians (JRT and BRT), medical students (DC, TA, XQ), and nursing staff (SU). During analysis, we reflected as a team on our positions and background and how they might impact the coding and interpretation of the data. Both EM physicians have formal training in research and qualitative methods and are bilingual certified providers (English/Spanish). SU is an ED nurse and a certified health care interpreter in Japanese. Medical students and nursing staff also had a 4-hour training session by primary investigators on qualitative methods and research approaches. DC and XQ conducted all interviews. Among the research team, 2 identify as female, 3 as male and 1 chose not to disclose their gender.

## Reporting standards

All results are reported according to the COREQ standards for reporting of qualitative research [22].

## Results

A total of 36 interviews were conducted with registered nurses (n = 29), nurse assistants (n = 3), and nurse practitioners (n = 4). Years of experience ranged from 0.5 to 36 years. Twenty-two of the participants spoke a language in addition to English. Barriers and facilitators were identified in all three of the BCW categories: capability, opportunity and motivation.

## Capability

Capability is comprised of physical and psychological components. Physical capability describes the provider's physical capacity to call or successfully interact with an interpreter [18]. Barriers to physical capability included the inability for both providers and patients to hear the remote interpreter because of ambient noise in the ED, especially with patients who are hard of hearing.

> *"Sometime over the phone it's difficult to hear. They have a hard time, the translator has a hard time hearing the patient, especially if it's like in our open areas. Have a lot of background noise."* Participant 9, RN, OVMC.

Participants also described the inherent difficulty of wearing personal protective equipment while using a phone. Other physical capability barriers included lack of patient privacy because of crowded physical spaces and lack of the time necessary to contact remote interpreter services in acute situations. The proposed interventions, based on the BCW intervention function of enablement, include increased availability of in-person interpreters and the designation of a quiet and private area where remote interpreter services can be used. (See Table 1) Facilitators related to physical capability included the presence of in-person interpreters and VRI devices where patients could see the interpreter's facial expressions and lips. Participants appreciated the ability of the interpreter to see and be seen by the patient, participate in nonverbal communication, which fostered a more personal connection with the patient.

*"If there's a lot of people in the room, it's hard to hear the interpreter on the phone…recently, a few months ago, we got the interpreters, physical bodies that go in with the doctors, and they do a good job."* Participant 14, RN, OVMC.

Psychological capability is the capacity to engage in the necessary thought processes, including comprehension, and reasoning to call or interact with an interpreter [18]. Participants were aware that language services were available, however many lacked training on the variety of ways to contact language services.

*"I would say a majority of the people don't even know they can use the phone… It's not well known. So, when I tell them to use the phone, they're like, oh, I didn't know I could do that."* Participant 31, RN, RRMC.

The proposed intervention was additional training for nursing staff on how to access language services. An additional barrier within the category of psychological capability was the inability to read discharge instructions written by providers in Spanish. The proposed intervention functions included enablement and training. Nurses could be enabled to overcome this barrier by augmenting the available prewritten bilingual patient information sheets that can be chosen by providers in the Electronic Health Record (EHR). Additional training for both providers and nurses on the policy and procedure for discharging patients with NELP is important given that providers should not be entering discharge instructions in non-English languages without the presence of the English translation.

Awareness of legal consequences was mentioned in the context of obtaining informed consent and was considered a facilitator for accessing language services in the category of psychological capability

*"But as soon as they start, you know, explaining procedures or like, hey, we need consent. That's where I draw the line. And I'm like, hey, you have to get a translator for this because I'm not trying to open myself to any legality or any issues like that."* Participant 34, RN, RRMC.

Self-awareness of the limitations of one's own non-English language skills was also a facilitator for contacting language services.

## Opportunity

The category of opportunity is divided into physical and social opportunity. The physical opportunity to call for an interpreter is made of the physical circumstances afforded by the environment [18]. Physical opportunity related barriers included the absence of working phones and the inability to find the iPad to utilize interpreter services.

*"We just lose track of them (iPads on wheels). There's been times where we are paging overhead for the iPads because nobody can find them. And we only have three, which I don't think is enough."* Participant 30, RN, RRMC.

**Table 1. Capability related barriers and facilitators.**

| Barrier/ Facilitator | BCW Source | Barrier/ Facilitator | *Quote (s)* | Intervention Function: Proposed Intervention (for barriers) |
|---|---|---|---|---|
| **Barriers** | | | | |
| | Capability (Physical) | Limitations of remote interpretation | *"(With) the person on the phone, you have to describe what you're talking about…and the patient has to yell to be heard. And sometimes they're on oxygen and they're short of breath. It's hard for them to yell."* Participant 14, RN, OVMC | **Enablement:** Increase availability of in-person interpreters and video remote interpretation devices. **Enablement:** Designate quiet, private areas for use with remote interpreters. |
| | Capability (Physical) | Patient acuity | *"There is not enough time to use (remote) translator services like the in-person translator is actually really good at dealing with patients who are sicker when we need information quickly."* Participant 10, RN, OVMC | **Enablement:** Increase availability of in-person interpreters in the ED. |
| | Capability (Physical) | Patient privacy | *"It's a big problem. There's zero privacy. It's hard enough to get a private room for proper assessment …we literally just triage out in the open. And when we do need translation services... Everybody can hear."* Participant 29, RN, RRMC | **Enablement:** Designate quiet, private areas for use with remote interpreter devices. |
| | Capability (Psychological) | Lack of training on available services | *"We also have a service through our phones, but I don't think we utilize them as much as we should…based on lack of knowledge on how to access the interpreter services."* Participant 30, RN, RRMC | **Training:** Provide detailed skills training on how to access language services. |
| | Capability (Psychological) | Discharge instructions not bilingual | *"(Regarding discharge instructions) Sometimes the doctors will do either or- just do an English, and so the patient can't understand it, or they'll do it in just Spanish. And so then I can't read what it is to tell the patient...how can I discharge the patient? I can't even read what I'm supposed to do."* Participant 28, RN, RRMC | **Enablement:** Augment prewritten bilingual discharge information so both nursing and the patient can read their content. **Training:** Train nurses on recommended procedure for discharge of patients with NEPLs. |
| **Facilitators** | | | | |
| | Capability (Physical) | Video format | *"They (patients) seem to be able to grasp things better and to communicate a little bit more seamlessly than if it's just the voice. If it's something to do with body language or mannerisms or whatnot, they seem a lot more in tune with the interpreter… it also works better for ASL for the sign language."* Participant 30, RN, RRMC | |
| | Capability (Psychological) | Awareness of legal consequences | *"A couple of years ago, they told us that we should not translate for a patient being consented for surgery or procedure, because of the legal aspect of it… because if it goes to court, they're going to question my abilities or background on who qualified me to interpret for a procedure.* Participant 14, RN, OVMC | |
| | Capability (Psychological) | Awareness of limitations | *"The doctors are sometimes very, very technical. And then we can't necessarily translate all the technicality for the patient. So that's where I would say, I (as a bilingual staff member) sometimes struggle even in Spanish."* Participant 1, NA, OVMC | |

RN = Registered Nurse, NA = Nurse Assistant, OVMC = Olive View-UCLA Medical Center, RRMC = Ronald Regan-UCLA Medical Center, NEPL = Non-English Preferred Language

The need to locate equipment contributed to the perception that using an interpreter would be slower or less efficient than an ad-hoc or in-person interpreter.

*"The E.R. is just a very dynamic place where sometimes time is of the essence. And to get a translator on the phone, and, you know, that could take, 5 to 10 minutes trying to look for a phone or iPad or call somebody."* Participant 34, RN, RRMC.

Proposed interventions, based on the intervention function of environmental restructuring, included designating a staff member to check the equipment every shift and a designated "parking space" for the VRI machines to eliminate the issues of nonfunctioning, uncharged or missing equipment. (See Table 2) In addition, reinforcement of training on the variety of methods to contact language services and how to escalate requests when problems are encountered would also be helpful. Facilitators included the availability of a phone at every bedside and the recent addition of in-person interpreters stationed in the ED at OVMC.

Social opportunity describes the "cultural milieu that dictates how a person perceives and thinks" [18]. Respondents described that it was socially acceptable for a provider to ask a colleague to interpret or use other ad-hoc interpreters such as family instead of using a certified interpreter.

*"I've been in some situations where the doc then pokes his head out of the curtain or room and says, "Is there anybody that can help me (interpret) in person?" That's that seems to be the fallback."* Participant 9, RN, OVMC.

Of note, this practice also poses unnecessary burden to some staff members. Multilingual nurses, especially those who speak Spanish, described an increased burden, as they were expected to both interpret for their colleagues and still perform the same amount of work.

*"I want to help, but we're task saturated. So when you get…'hey, can you just discharge this patient because they only speak Spanish?'... Like, you want to help out your colleague, but you're like, please don't pull me for this stuff."* Participant 32, RN, RRMC.

The proposed intervention, based on the intervention function of modelling, was to identify senior members of the physician and nursing staff to model the appropriate use of language services and disseminate information about language services in real time when situations arise where staff are relying on untrained bilingual staff. Social opportunity also served as a facilitator to language access at OVMC where it was made clear that the administration expected staff to call for language services when necessary.

## Motivation

Motivation is comprised of automatic and reflective processes. Automatic motivation refers to automatic processes involving emotions and impulses that arise from associative learning and/or innate dispositions [18]. This can include habitual processes and emotional responses in decision making. Automatic motivation was an identified barrier to calling an interpreter as some respondents had an established habit of using ad hoc interpreters.

*"There's always people around here. I mean, I know we're not supposed to, but If I'm in triage I'll even grab a housekeeper for a quick moment (to interpret)."* Participant 18, RN, OVMC.

Participants described aiming to just "get by" when interacting with patients with NELPs using workarounds including the provider's own non-fluent language skills, Google translate, or ad-hoc interpreters (e.g., family members, colleagues, housekeeping staff).

**Table 2. Opportunity related barriers and facilitators.**

| Barrier/ Facilitator | BWC Source | Barrier/ Facilitator | Quote (s) | Intervention Function and Proposed Intervention Intervention (for barriers) |
|---|---|---|---|---|
| **Barriers** | | | | |
| | Opportunity (Physical) | Broken/ missing equipment | "Sometimes they (phones) go missing. Sometimes the wires are torn. Sometimes we've just wiped them down so much with cloths that they're just gunked up." Participant 20, RN, OVMC | **Environmental Restructuring:** Designate staff to check equipment regularly. **Environmental restructuring:** Establish a designated "parking space" for video remote interpreter devices. |
| | Opportunity (Physical) | Time pressure | "The E.R. is just a very dynamic place where time is of the essence. To get a translator on the phone- that could take 5 to 10 minutes trying to look for a phone or iPad or call somebody. So, I get why sometimes shortcuts are taken." Participant 34, RN, RRMC "If there are patients that have very specific dialects or maybe languages we don't use as often… there was a patient that Cambodian was his preferred language. And we were waiting for an extended periods of time for an interpreter." Participant 35, RN, RRMC | **Environmental Restructuring:** Prioritize in-person interpreters for acute, time-sensitive issues. **Training:** Train staff on the various options for accessing interpreters for languages of lesser diffusion. |
| | Opportunity (Social) | Culture of asking colleagues/staff to interpret | "Most providers will try their best to do a mixture of Spanglish to get what we need done. And (they) just go ahead and render care. If I'm at the bedside and the doctor doesn't really know Spanish, they'll look at me. They're like, hey, you know Spanish, right? I'm like, Yeah, sure. We're here at the bedside. I'm not going to make it awkward. I'm going to translate for you." Participant 34, RN, RRMC | **Modelling:** Identify senior members of physician and nursing staff as champions to model use of language resources and disseminate information on methods for contacting language services. |
| **Facilitators** | | | | |
| | Opportunity (Physical) | Augmenting available equipment | "They put phones in every room. We didn't have that before. I know they care. I mean, I see that the administration or DHS cares about that (language access). Participant 4, NP, OVMC | |
| | Opportunity (Physical) | Availability of in-person interpreters in the ED | "The in-person translator is actually really good at dealing with patients who are sicker and then we need like information quickly." Participant 10, RN, OVMC | |
| | Opportunity (Social) | Administration expects staff to use interpreters | "They always tell us you're not, you know, if you have a patient that does not speak your language, yes, use the translator. It does take some time. Therefore, to use it so the person can understand what's going on." Participant 6, RN, OVMC | |

RN = Registered Nurse, NP = Nurse Practitioner, OVMC = Olive View-UCLA Medical Center, RRMC = Ronald Regan-UCLA Medical Center

Further, participants reported that Spanish is treated differently than other languages. The frequency of contact with the Spanish language made providers feel they could communicate enough without calling an interpreter even when not proficient or certified, potentially leading to less language access for Spanish-speaking patients.

"They tried to do like a broken Spanish… there's a gap already in communication, let alone like a broken Spanish. So I think that among my colleagues, they probably should use it (language services) more than what they do." Participant 3, RN, OVMC

Reflective motivation refers to the processes that include evaluations and plans for how a provider uses language services [18]. Reflective motivation facilitated contacting interpreters when providers connected language services to ideas of patient's rights and equitable provision of care.

*"The care that an affluent English-speaking individual versus a Spanish guy gets for his chest pain is very different… we just have to do our part and get translation services."* Participant 34, RN, RRMC.

Other providers focused on the positive impact of proper communication on patient care, outweighing any potential time costs associated with calling an interpreter.

*"The care I am giving should not be substandard. I use the language services, and I'm happy with that, and I don't mind if I see one patient less."* Participant 16, NP, OVMC.

Participants who prioritized using language services countered perceptions of an efficiency cost by focusing on the benefit to patient outcomes and satisfaction. They also identified potential communication costs associated with forgoing interpretation. Further, recognizing language access as an issue of patient autonomy and recognizing potentially dangerous miscommunications and how they occurred also facilitated willingness to engage interpreters. (See Table 3) Encouraging reflective exercises during meetings and trainings are viable interventions to promote this type of reflection.

## Discussion

Previous studies have shown that patient outcomes and satisfaction suffer when providers do not use professional interpreters when communicating with patients with a NELP [14,23–25]. Professional, as compared to ad hoc, interpreters have advanced skills that leads to more accurate interpretation [26]. Resource utilization arguments also support the use of professional interpreters [27], however, in most health care settings, underutilization or "getting by" remains the norm [28,29]. Emergency nursing is particularly challenging for communication with patients with a NELP because it is comprised of many short but important communications with undifferentiated patients. Through use of the BCW, qualitative interviewing of nursing staff elucidated concrete and actionable information about barriers and facilitators to accessing interpreters for ED patients with a NELP. The information obtained was mapped to the corresponding BCW intervention functions to design potential interventions with greatest likelihood of improving rates of interpreted conversations between nursing staff and patients with a NELP. The important next step in this research will be a formal implementation of the proposed interventions with measurement of process and patient outcomes.

We found that one of the key lessons to be learned from this study is the extent to which administrative decisions and interventions have the potential to positively impact daily practice. The underuse of interpreters is sometimes considered an engrained culture that is difficult to change, but participants in this study illuminated potential concrete administrative fixes that are promising to improve rates of interpreted conversations. Administrators should recognize the unique and fast paced environment of the ED and allocate more in person and video resources to the area because in-person interpreters, especially, have more flexibility to pivot with the team as acute situations evolve. Devoting more in-person resources that can handle complex patients (time critical, hypoxic patients with multiple team members in the room and high levels of ambient noise) may influence the calculus of time tradeoffs described by staff that led to ad hoc interpreter use. This preference for a visual connection also extended to the preferred methods of remote interpretation. Participants stated they felt the video-based interpreters (i.e. iPad) were better able to connect with patients, and patients felt they were speaking to a person as compared to phone interpretation. This is concordant with other studies showing a preference for this interpretation medium [30].

**Table 3. Motivation related barriers and facilitators.**

| Barrier/ Facilitator | BCW Source | Barrier/ Facilitator | *Quote* | Intervention Function: Proposed Intervention (for barriers) |
|---|---|---|---|---|
| **Barrier** | | | | |
| | Motivation (Automatic) | Calling an interpreter is outside the provider's usual routine. | *"We've done it so much… a lot of patients come in with similar symptoms, if you need to go more in detail, and we don't know how to, you know, ask a question, then we would utilize those (language access) services, but for the most part, I think we do kind of rely on our own knowledge."* Participant 10, RN, OVMC *"I use the phone as a last resort, I'd rather get a person. But there's always people around here. I mean, I know we're not supposed to, but if I'm in triage I'll grab even a housekeeper for a quick, quick moment."* Participant 18, RN, OVMC *"I try to use my phone, like Google Translate, for example, there's a lot of the patients that are Armenian, I always try to go find a clerk or my boss that speaks Armenian."* Participant 19, NA, OVMC | **Modelling:** Identify senior members of nursing staff as champions to model use of language resources and disseminate information on methods for contacting language services. |
| | Motivation (Automatic) | Perception of time trade off | *"It just takes too long. Yeah you know, we're supposed to use it. I don't use it that often. That's my last resort. Like last resort."* Participant 4, NP, OVMC | **Enablement:** Make devices easier to locate and remote interpreter services faster to access. |
| **Facilitator** | | | | |
| | Motivation (Reflective) | Desire to provide equitable care | *"We should know better and know to offer all services to all our patients, regardless of what their needs may be, to provide fair, just healthcare. Right? Like get a translator on the phone, speak to them in their native language. Take the time to translate things and explain things as you would to somebody that speaks English."* Participant 34, RN, RRMC | |
| | Motivation (Reflective) | Quality considerations | *"If I have the time, I love to do it (use an interpreter). Because I think it makes the patient feel more comfortable expressing themselves. And they're more satisfied, they don't feel rushed so much. Or like they miss something. Um, but I consider myself that, you know, it's good for me, because I feel like I can understand everything they want to tell me, I'm not missing anything."* Participant 14, RN, OVMC | |
| | Motivation (Reflective) | Patient autonomy | *"The patient doesn't really know what's going on with themselves, how do you expect them to have good outcomes and be able to take control of their health care. You can't because they don't really grasp what's going on."* Participant 34, RN, RRMC *"I've witnessed that with some doctors where they attempt to talk to a patient in their Spanish and I can hear that their Spanish is not very good. And I usually jump in, and I give them the translator phone. And because a lot of times that patient, you know, they're especially with my culture they're embarrassed to tell the doctor, hey, can you use a translator because your Spanish is not that great. They don't want to say something."* Participant 29, RN, RRMC | |
| | Motivation (Reflective) | Recognizing dangerous miscommunications | *There was a patient yesterday that came to the front…He wanted a physical and he spoke Spanish, and the person at the front doesn't speak Spanish, So it's like, okay, what's wrong with you? And he ended up like pointing to his legs, I think they were swollen. He gets signed in for leg pain…Long story short, his H&H was like super low (from hematemesis) …But he was not leg pain… there was more to it... Participant 12, RN, OVMC* | |

RN = Registered Nurse, NA = Nurse Assistant, NP = Nurse Practitioner, OVMC = Olive View-UCLA Medical Center, RRMC = Ronald Regan-UCLA Medical Center.

The second key point of the results are that consideration of the environment and workflows when constructing interventions have been critical. Having a dedicated parking space for the VRI machine, for instance, so that staff does not waste time searching for it, is a remarkably simple intervention but was mentioned by participants as an important need to influence nursing staff's willingness to use the technology. Prior interventions that focused on enablement were mentioned as examples of useful interventions (having bedside phones in each room that autodial the interpreter line).

Finally, prior training and education was mentioned as influencing nursing staff decisions regarding language access. Those staff who had been exposed to education on hospital policy and the legal basis for the right to language access cited this training as influencing their thinking and empowering them to call interpreters, especially in the setting of informed consent. This is similar to previous studies that when organizational investment in language services is apparent providers prioritize language access [31]. Developing this element of workplace culture, too, may increase provider willingness to call an interpreter [32]. Finding staff who are "champions" of promoting and using these services would serve to model this behavior and has been effective [33]. Training staff champions to demonstrate and promote proper language access within clinical areas would serve to promote institutional knowledge, serve to promote social opportunity, and further advocate for the implementation of language services in common provider patient interactions in the ED.

A secondary benefit to the administrative investments described above is reduced burden for bilingual staff members. Bilingual nursing staff described a cost to their work performance when they were pulled as an ad-hoc interpreter for nonemergent conversations.

This has similarly been described by nurses in other settings and noted to be a sort of invisible additional workload that adds up [34]. Chang et al studies this phenomenon of bilingual staff being "pulled away" and found it was driven by the assumption that all bilinguals could interpret medical terminology and linked it to a workplace culture that did not address risk related to language barriers [35]. Utilizing the interventions proposed above will not only raise rates of interpreted conversations but also empower overburdened bilingual staff to remind colleagues to access language services.

Further study is needed to examine each of these interventions and their impact. Possible outcomes to examine may include documented interpreter usage rates, patient satisfaction ratings, and identifying concordance between patient's perceived content of conversations and providers.

## Limitations

The study was conducted at two urban EDs in Los Angeles and therefore may not represent all settings. Given the high proportion of patients with NELPs and bilingual providers in Los Angeles, staff in this setting may generally have had more exposure to the topic of communication with NELP patients than in other settings. This study did not collect the total number of staff invited to participate or the proportion of those who agreed which makes participation bias difficult to assess in this study. Because NPs and NAs are present at only one of the two sites, sampling of those roles was limited, but they were included as important stakeholders whose views may differ from the RNs.

## Conclusions

The ED is a setting where fast and accurate communication is critical. We identified barriers to and facilitators of ED nursing utilization of language access related to each construct of the BCW. Analysis of nursing perspectives led to a variety of concrete and actionable interventions that ED and hospital administrators can implement to increase rates of interpreted ED nursing encounters. Addressing the underuse of interpreters through multifaceted interventions within the healthcare system has the potential to support lasting positive change for communication with patients with a NELP and their subsequent health outcomes.

## Supporting information

**S1 File. Interview guide.**
(DOCX)

## Acknowledgments

**Prior presentations:** Results have been presented in part at the Society for Academic Emergency Medicine May 17, 2022 and the Emergency Nursing 2024 Conference September 4, 2024.

## Author contributions

**Conceptualization:** Daniel Cordova, Jesus R. Torres, Breena R. Taira.

**Data curation:** Daniel Cordova, Jesus R. Torres, Shiou Udagawa, Xin Qi, Tucker Avra, Breena R. Taira.

**Formal analysis:** Daniel Cordova, Jesus R. Torres, Shiou Udagawa, Breena R. Taira.

**Funding acquisition:** Jesus R. Torres, Breena R. Taira.

**Investigation:** Daniel Cordova, Jesus R. Torres, Shiou Udagawa, Xin Qi, Tucker Avra, Breena R. Taira.

**Methodology:** Daniel Cordova, Jesus R. Torres, Breena R. Taira.

**Project administration:** Daniel Cordova, Jesus R. Torres, Shiou Udagawa, Breena R. Taira.

**Resources:** Daniel Cordova, Jesus R. Torres, Shiou Udagawa, Breena R. Taira.

**Software:** Jesus R. Torres, Breena R. Taira.

**Supervision:** Jesus R. Torres, Breena R. Taira.

**Writing – original draft:** Daniel Cordova, Jesus R. Torres, Breena R. Taira.

**Writing – review & editing:** Daniel Cordova, Jesus R. Torres, Breena R. Taira.

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
