## [Decision Letter · Decision Letter 0]

29 May 2025

Dear Dr. Cordova,

Thank you for submitting your manuscript to PLOS ONE. After careful consideration, we feel that it has merit but does not fully meet PLOS ONE’s publication criteria as it currently stands. Therefore, we invite you to submit a revised version of the manuscript that addresses the points raised during the review process.

We look forward to receiving your revised manuscript.

Kind regards,

Meryem Merve Ören Çelik

Academic Editor

PLOS ONE

4. Please include your tables as part of your main manuscript and remove the individual files. Please note that supplementary tables (should remain/ be uploaded) as separate "supporting information" files

Reviewers' comments:

Reviewer's Responses to Questions

**Comments to the Author**

1. Is the manuscript technically sound, and do the data support the conclusions?

Reviewer #1: Yes

Reviewer #2: Yes

2. Has the statistical analysis been performed appropriately and rigorously?

Reviewer #1: Yes

Reviewer #2: Yes

3. Have the authors made all data underlying the findings in their manuscript fully available?

Reviewer #1: Yes

Reviewer #2: Yes

4. Is the manuscript presented in an intelligible fashion and written in standard English?

Reviewer #1: Yes

Reviewer #2: Yes

Reviewer #1: This qualitative study of 36 Emergency Department nurses elucidated many potentially surmountable barriers to using interpreters for NELP patients. The reference to the author’s website was probably superfluous, but otherwise the content was appropriate. One of the areas that could have benefited the reader more to investigate was how nurses felt comfortable proceeding with broken Spanish. It also would have been nice to know the split of languages other than English preferred by the patients.

Reviewer #2: The study provides important insights into real-world barriers and facilitators to interpreter use in fast-paced clinical settings. The findings are well supported by quotes, and the proposed interventions are concrete and practical. I especially appreciate that the authors also address the burden placed on bilingual staff, which is an often overlooked issue.

I recommend minor revisions before acceptance. Please consider the following:

Response rate: The number of participants is clearly reported, but the manuscript does not mention how many staff were invited to participate. Adding this information would help readers assess possible participation bias.

COREQ checklist – minor gaps:

While the authors refer to using COREQ standards, some points could be clarified. For example, it is not clear whether participants reviewed their transcripts or the study findings.

The interview guide is mentioned but not shared. Providing a brief summary or attaching it as supplementary material would improve transparency.

Future implementation:

The manuscript outlines valuable interventions. It would be helpful to briefly mention how the impact of these interventions might be measured in future work (e.g., interpreter usage rates, patient feedback).

Champion staff model:

The idea of having staff “champions” for language access is strong. Still, a brief note on how this approach could be supported or sustained would make the recommendation more complete.

Overall, this is a thoughtful and useful contribution that I believe will be of interest to both researchers and healthcare practitioners. I support publication after minor revisions.

**Do you want your identity to be public for this peer review?** For information about this choice, including consent withdrawal, please see our Privacy Policy

Reviewer #1: **Yes: ** Jay M. Brenner

Reviewer #2: **Yes: ** Buğra Taygun Gülle

---

## [Author Response · Author response to Decision Letter 1]

5 Aug 2025

Dear reviewers,

Please find attached our revised manuscript, "Nursing Perspectives on Advancing Language Access in the Emergency Department: A Qualitative Study," submitted for consideration in PLOS ONE. We thank the reviewers for their insightful feedback. We have addressed all requested changes and have listed our response below as requested and believe the manuscript is significantly improved. We are available to provide any further revisions or information if needed.

Please see the list of requested changes and author responses below:

• The requested changes have been made and the PLOS ONE’s style requirements to the best knowledge of the study team. If further changes are required the team is happy to adjust further.

• Not applicable, as this is a qualitative study. Also, providing the full qualitative transcripts will increase the risk that the participant could be identified.

• The requested statement has been added to the methods section.

4. Please include your tables as part of your main manuscript and remove the individual files. Please note that supplementary tables (should remain/ be uploaded) as separate "supporting information" files

• The tables have been added to the end of the main manuscript.

• All references are correct to the knowledge of the study team.

Reviewers' comments:

Reviewer's Responses to Questions

Comments to the Author

Reviewer #1: This qualitative study of 36 Emergency Department nurses elucidated many potentially surmountable barriers to using interpreters for NELP patients. The reference to the author’s website was probably superfluous, but otherwise the content was appropriate. One of the areas that could have benefited the reader more to investigate was how nurses felt comfortable proceeding with broken Spanish. It also would have been nice to know the split of languages other than English preferred by the patients.

• We thank the reviewers for the suggestion but we unfortunately do not have that information at this time.

Reviewer #2: The study provides important insights into real-world barriers and facilitators to interpreter use in fast-paced clinical settings. The findings are well supported by quotes, and the proposed interventions are concrete and practical. I especially appreciate that the authors also address the burden placed on bilingual staff, which is an often overlooked issue.

I recommend minor revisions before acceptance. Please consider the following:

Response rate: The number of participants is clearly reported, but the manuscript does not mention how many staff were invited to participate. Adding this information would help readers assess possible participation bias.

• We did not collect information on total number of nurses approached and those that agreed to participate. A brief statement regarding this has been added to the limitations portion of the manuscript.

COREQ checklist – minor gaps:

While the authors refer to using COREQ standards, some points could be clarified. For example, it is not clear whether participants reviewed their transcripts or the study findings.

• A statement regarding whether participants reviewed their transcripts was added to the manuscript.

The interview guide is mentioned but not shared. Providing a brief summary or attaching it as supplementary material would improve transparency.

• The study guide was added to the supplementary material.

Future implementation:

The manuscript outlines valuable interventions. It would be helpful to briefly mention how the impact of these interventions might be measured in future work (e.g., interpreter usage rates, patient feedback).

• A statement regarding how the impact of these interventions may be measured in the future was added to the manuscript

Champion staff model:

The idea of having staff “champions” for language access is strong. Still, a brief note on how this approach could be supported or sustained would make the recommendation more complete.

• A brief statement regarding the approach of using staff “champions” has been added to the manuscript.

6. PLOS authors have the option to publish the peer review history of their article (what does this mean?). If published, this will include your full peer review and any attached files.

• We as a group would prefer to not publish the peer review history.

Thank you very much for your consideration.

Sincerely,

Daniel Cordova, MD

---

## [Decision Letter · Decision Letter 1]

27 Oct 2025

Nursing Perspectives on Advancing Language Access in the Emergency Department: A Qualitative Study

PONE-D-24-46814R1

Dear Dr. Cordova,

We’re pleased to inform you that your manuscript has been judged scientifically suitable for publication and will be formally accepted for publication once it meets all outstanding technical requirements.

Kind regards,

Stephen R. Milford

Academic Editor

PLOS ONE

Additional Editor Comments (optional): 

Please note that I have only recently been assigned the role of academic editor for this paper (within the last few days). An academic editorial re-assignment can take place for a number of reasons (editors resign, fall ill etc.) and this invariably delays the process. A new academic editor needs to familiarise themselves with the paper and the reviews/revisions conducted. I have tried to look over the history of this paper. It appears that there was an effort to have the revisions of the paper reviewed by both original reviewers. However, only one of the original reviewers seems to have submitted an acceptance while the second reviewer has not submitted a review of the revisions requested. Considering the second reviewer originally recommended minor edits, and this paper has been submitted almost a year ago, I feel that it is appropriate to issue an acceptance decision at this stage.

I appreciate that this process has been lengthy. While I am not in a position to speak for the journal, I appreciate your patience and hope that you are happy with this decision.

I wish you all the best.

Reviewers' comments:

Reviewer's Responses to Questions

**Comments to the Author**

Reviewer #1: All comments have been addressed

2. Is the manuscript technically sound, and do the data support the conclusions?

Reviewer #1: Yes

3. Has the statistical analysis been performed appropriately and rigorously?

Reviewer #1: Yes

4. Have the authors made all data underlying the findings in their manuscript fully available?

Reviewer #1: Yes

5. Is the manuscript presented in an intelligible fashion and written in standard English?

Reviewer #1: Yes

Reviewer #1: I appreciate the revisions. I recommend acceptance for publication. I especially like the study's conclusions.

**Do you want your identity to be public for this peer review?** For information about this choice, including consent withdrawal, please see our Privacy Policy

Reviewer #1: No

---

## [Editor Report · Acceptance letter]

PONE-D-24-46814R1

PLOS ONE

Dear Dr. Cordova,

I'm pleased to inform you that your manuscript has been deemed suitable for publication in PLOS ONE. Congratulations! Your manuscript is now being handed over to our production team.

Kind regards,

on behalf of

Dr. Stephen R. Milford

Academic Editor

PLOS ONE